# Investigation of the Performance of Various Low-Cost Radon Monitors under Variable Environmental Conditions

**DOI:** 10.3390/s24061836

**Published:** 2024-03-13

**Authors:** Daniel Rábago, Enrique Fernández, Santiago Celaya, Ismael Fuente, Alicia Fernández, Jorge Quindós, Raúl Rodriguez, Luis Quindós, Carlos Sainz

**Affiliations:** 1Laboratory of Environmental Radioactivity (LaRUC), University of Cantabria, 39011 Santander, Spain; daniel.rabago@unican.es (D.R.); enrique.fernandez@unican.es (E.F.); celayas@unican.es (S.C.); alicia.fernandezv@unican.es (A.F.); jorge.quindos@unican.es (J.Q.); luis.quindos@unican.es (L.Q.); sainzc@unican.es (C.S.); 2Siglo 21 Consultores S.L., Av. Rosalía Castro, 94, Perillo, 15172 A Coruña, Spain; raul.rodriguezal@alumnos.unican.es

**Keywords:** response time, intercomparison, low-cost radon monitor

## Abstract

A comparison of low-cost radon monitors was conducted at the Laboratory of Natural Radiation (LNR). The monitors we evaluated were EcoQube, RadonEye, RadonEye Plus2, Spirit, ViewPlus, ViewRadon and WavePlus. An AlphaGUARD monitor calibrated at the Laboratory of Environmental Radioactivity of the University of Cantabria (LaRUC), accredited for testing and calibration according to ISO/IEC 17025, provided the reference value of radon concentration. The temporal stability of the monitors was studied, obtaining a percentage of missing records ranged from 1% to 19% of the data. The main technical characteristics studied were temporal stability, measurement ranges, accuracy, correlation and response time. The main results show that the measurement ranges align with those specified by their manufacturers, with percentage differences with respect to the reference monitor of between 5% and 16%. The diversity found for response time is remarkable, with values ranging from 1 to 15 h, with Pearson correlation factors between 0.63 and 0.90.

## 1. Introduction

The presence of the radioactive gas radon (^222^Rn) in residential areas and workplaces has gained significant attention in public health and occupational safety, due to its potential risks to human health. The inhalation of radon and its progeny has been associated with an increased risk of developing lung diseases, especially lung cancer, making it an issue of global relevance [1]. The European Directive 2013/59/EURATOM [2], which sets out guidelines and measures to protect the population from radon exposure, even at low concentration levels, addresses the importance of addressing this issue.

To effectively implement these measures, radon measurement with low-cost devices has emerged as a practical, valuable possibility for a wide range of users. Accurate assessment of radon exposure is essential to adopt, if necessary, appropriate mitigation strategies and reduce the associated risks. This type of device is presented as a practical and economical alternative, allowing continuous monitoring of radon concentration in ambient air.

In this context, we can find active and passive techniques for radon measurements. The first one involves devices that directly record radon concentration at predefined time intervals, while the second one uses detectors that are subsequently analysed in the laboratory, providing a single average value of exposure or concentration [3,4]. The choice between the two options depends on several factors, such as the length of the measurement period, the level of accuracy required and/or the budget available for the measurements.

Quality control in radon measurement is of utmost importance to ensure reliable results. Quality assurance activities, such as intercomparison between different devices, calibration in radon chambers, or comparison of controlled versus variable conditions, are crucial elements in this process [5,6]. Initiatives such as the European metrological project Metrology for Radon Monitoring (MetroRADON) have emerged as pillars to establish standards and protocols for radon measurement, providing a solid basis for data validation and comparison [7].

The main objective of this study is to evaluate the stability, accuracy and response time of low-cost commercial devices with different technical characteristics under actual and demanding conditions where radon concentrations can be extremely high and variable, depending on environmental conditions.

## 2. Materials and Methods

### 2.1. Place of Study: Laboratory of Natural Radiation LNR

The comparison of radon monitors was carried out at the Laboratory of Natural Radiation (LNR) (see Figure 1), located at the former uranium mine managed by ENUSA Industrias Avanzadas (Saelices el Chico, Salamanca, Spain) [8]. This facility has been used since 2000 to carry out several intercomparison exercises and in situ calibrations, addressing the measurement of radon concentration in indoor air, radon exhalation rate from soil, radon concentration in soil, and external gamma radiation dose rate under environmental conditions [9,10,11,12]. The suitability of this environment for such activities is attributed to the high content of radioactive elements present in the soil, together with the site-specific environmental conditions.

In the LNR, there are several separate rooms used as radon chambers. In them, the radon source is the soil and the indoor concentration variations inside the building are conditioned by the environmental conditions. The study was conducted in Room1 (see Figure 1), which has a volume of approximately 45 m^3^. There are two windowless exterior walls, whose the only direct connection to the outside is provided by a pressurisation system installed in 2022, which was switched off during this study. The only connection to the general room of the LNR is through a metal door, which is well sealed. In addition, a system is in place to monitor the environmental conditions in Room1 and outside the LNR. A low power fan (13 W) was installed to homogenise the radon atmosphere. The device used to measure the environmental conditions in Room1 was a TESTO 176-P1 with a humidity and temperature probe. The mining facility’s weather station was used for the outdoor environmental conditions.

### 2.2. Selection and Preparation of Radon Monitors

A technical and market review was carried out to identify available radon monitors within the price range set as “low cost” (under EUR 1000). In selecting radon monitors, several factors were considered, such as accuracy of measurements, durability of the device, ease of use and data storage capacity. Concerning the latter characteristic, devices that did not store data or did not offer the possibility to download data for further processing were not considered. Therefore, although several monitors may meet the price condition, in this study the term low-cost should include the other features discussed above.

Most radon monitors currently on the market include the measurement of environmental parameters (humidity, atmospheric pressure, temperature). Some also include parameters related to air quality (CO_2_ concentration, volatile organic compounds VOC, particle size concentration, etc.). However, none of the abovementioned parameters were considered in this study, which focused exclusively on radon measurement.

Finally, seven radon monitors were selected, whose main technical characteristics related to radon are specified in Table 1. In all cases, the device operates without a pump, i.e., the air intake mechanism in the detection chamber is passive.

The technical characteristics of the AlphaGUARD radon monitor, which has been used as the standard that will provide the reference radon concentration at each instant in Room1 of the LNR, are shown in Table 1. This radon monitor has been calibrated in the calibration chamber of the Laboratory of Environmental Radioactivity of the University of Cantabria (LaRUC) [13,14,15], which is accredited according to ISO/IEC 17025 [16].

Before the start of the exposure, the radon background of each monitor has been determined, i.e., the average concentration provided by the device in the absence of radon. This is due to electronic noise or contamination of the detector by long-lived radionuclides [17]. This background value must be subtracted from each of the individual measurements, as it does not pertain to the actual radon level of the environment in which the monitor is exposed. The LaRUC calibration chamber was used to assess the background of each monitor by placing all monitors inside in the absence of radon sources. It was closed, ensuring the seal, and purged with radon-free air from a canister stored long enough to ensure the absence of radon.

The comparison was conducted in Room1 of LRN for 54 days, between 1 July 2023 and 23 August 2023. All monitors were placed in the middle of the room, on a table on a surface of 0.5 m by 0.5 m at 1 m high to ensure homogeneity of the radon concentration. The room remained completely enclosed, with no direct air ingress during the experiment.

### 2.3. Data Analysis

Once the exposure finished, we downloaded the raw data provided by each monitor. We obtained a series of correlative dates according to each monitor’s integration or measurement time and the radon *C** concentration in Bq/m^3^. As shown above, it is necessary to subtract the obtained background *B* from each of the measures. Similarly, to maintain traceability of measurements, applying the calibration factor or correction factor *F* provided by a calibration laboratory would be necessary. In this case, none of the monitors under study claim to be externally calibrated, so no correction was applied to the provided radon concentration. This way, the quality of the measurements provided from the factory-set data was assessed. In general, radon concentration *C* measurements should be corrected for:(1)$$C=\left(C^{*}-B\right)\times F$$
where *C** is the raw radon concentration (Bq/m^3^) provided by the monitor, *B* is the background (Bq/m^3^), and *F* is the calibration factor (dimensionless) given by a calibration laboratory with traceability to national or international standards.

In the case of the AlphaGUARD reference monitor, the calibration factor or correction factor provided by the LaRUC is *F_ref_* = 1.00 ± 0.09 (k = 2). This factor was obtained during a period of stability of the radon concentration in a radon chamber, by comparing a standard instrument, calibrated in a laboratory on a metrological scale above, and the radon concentration measured by the instrument to be calibrated (AlphaGUARD). Its uncertainty was obtained from the law of propagation of uncertainty [18], taking into account the uncertainty of the radon concentration measured by the standard, the uncertainty of its calibration factor, the statistical uncertainty of the net radon concentration measured by the instrument to be calibrated, and its background uncertainty.

To analyse the temporal stability of the devices, they were kept exposed to known concentrations. With this monitoring, we obtained the periods in which the sensors did not register the data correctly.

Accuracy, response time, and correlation between series and measurement ranges were also evaluated. The accuracy study compares the measurements provided by the monitor and the reference device. The percentage difference *D* (%) from the reference in a given range or for individual values can be obtained as follows:(2)$$D\left(\%\right)=100\times \frac{\left(C-C_{ref}\right)}{C_{ref}}$$
where *C* is the radon concentration measured by the monitor, and *C_ref_* is the concentration measured by the reference device. According to Equation (1), the above concentrations are corrected by their background *B* and their calibration factor *F*.

Once the differences between the series recorded by each monitor and the reference monitor were obtained, a comparison of means was performed using Student’s *t*-test [19]. This test was used to determine whether there was a significant difference between the means of two independent groups. The null hypothesis (H_0_) is formulated as there is no significant difference between the means and the alternative hypothesis (H_1_), in which there is a significant difference between the two series. A confidence level of α = 0.05 was used, representing the accepted probability of making a type I error, i.e., incorrectly rejecting the null hypothesis when it is true, in this case 5%. We compare the *p*-value obtained in the statistical test; if it is less than α, we conclude that there is sufficient evidence to reject the null hypothesis and affirm a significant difference between the means of both time series.

On the other hand, to study the relationship between the series of each monitor and that provided by the reference monitor, a correlation study was carried out using Pearson’s correlation coefficient *r* [20]. This coefficient takes values between −1 and 1. A value of 1 indicates a perfect positive correlation (variables increase simultaneously), a value of −1 indicates a perfect negative correlation (one variable decreases while the other increases), and 0 indicates no linear correlation. Intermediate values indicate a greater or lesser degree of relationship between the two series. To determine whether the observed correlation is statistically significant, we calculate the *p*-value associated with the correlation coefficient and take as the null hypothesis (H_0_) that there is no correlation between both time series, and as the alternative hypothesis (H_1_) that there is a correlation between both time series. Suppose the *p*-value is less than the confidence level α = 0.05. In that case, we reject the null hypothesis and conclude that there is a significant correlation between both time series at a confidence level of 95%.

Finally, the proposed method for estimating the response time of radon monitors is based on the references [13,21]. It consists of analysing the time it takes for each monitor to reach a percentage of the final radon concentration measured with the reference device in a given time interval. The proposed key percentages are 10%, 50% and 90%, chosen to be at the beginning, middle and end of each time interval.

## 3. Results and Discussion

Before exposure, the background *B* of the monitors was determined and subtracted from the corresponding individual raw data *C** of the radon concentration provided by the device. The results for the device’s backgrounds and the integration time (defined by default in each of them) used in the determination of the background and during the exposure period in the LNR are shown in Table 2. The standard deviation SD of the background has been calculated based on a Gaussian distribution analysis. It is observed that the background value of the monitors is in the range between 0 and 20 Bq/m^3^. In the case of the reference device, the background value is 103 Bq/m^3^; this is because there is a fixed device in Room1 of the LNR, which has been continuously subjected to high radon concentrations.

The evolution of radon concentration during the exposure period in Room1 of the LNR from 1 July 2023 to 23 August 2023 given by the reference device (AlphaGUARD) and the main environmental conditions (atmospheric pressure *P*, relative humidity *rH* and temperature *T*) inside the Room1 and outside are shown in Figure 2. A daily evolution pattern is observed, with maximum values found between 12:00 and 14:00. Generally, such daily variations are in the range of 500–1500 Bq/m^3^. However, several events with values ranging from 3000 to 15,000 Bq/m^3^ can be observed.

Table 3 shows descriptive statistics of the environmental conditions during the exposure period. Pearson’s correlation *r* was performed on a matrix of all the variables described. There is a negative correlation between radon concentration and indoor temperature (*r* = −0.32) and a positive correlation with atmospheric pressure (*r* = +0.18). Both indoor and outdoor atmospheric pressure are almost perfectly correlated (*r* = +0.99), there is an absolute difference between the two, which may be due to the difference in altitude between the weather station and LNR. In the case of the relationship between humidity and temperature, there is a negative correlation between the two variables, *r* = −0.9 and *r* = −0.4 for outdoor and indoor, respectively.

Also noteworthy is the time lag between indoor and outdoor temperatures, with the maximum outdoor temperatures generally occurring between 16:00 and 18:00 and the corresponding indoor temperatures between 3:00 and 5:00. The daily range of variation in the indoor case is notably lower, 1.5 °C compared to 15 °C in the outdoor case, approximately. This difference in absolute variation is also found in relative humidity, with average variations of 6% indoors compared to 60% outdoors.

The radon concentration measured by every device during the entire exposure period in Room1 is shown in Figure 3. This graphical representation has been obtained from the time series downloaded from each monitor, from which the corresponding background *B* shown in Table 2 has been subtracted according to Equation (1).

From the individual radon concentration measurements shown in Figure 3, each device’s average, the median, the maximum value reached and the interquartile range IQR, defined as the difference between the third quartile Q3 (75th percentile) and the first quartile Q1 (25th percentile), have been obtained. The study took place over a 54-day period. Each monitor has a specific integration time, which determines how frequently it takes measurements. For monitors with an integration time of 10 min (such as AlphaGUARD and Spirit), there should be a total of 7776 measurement intervals over the 54 days. For monitors measuring every 20 min (like ViewPlus and ViewRadon), there should be 3888 intervals, and for those measuring every 60 min (such as EcoQube, RadonEye, RadonEye Plus2, and WavePlus), there should be 1296 intervals. These expected numbers of measurement intervals were then compared with the actual number of data points recorded by each monitor. This comparison helps determine the percentage of data that was not captured, or “lost,” by each monitor during the study period. Finally, the percentage difference concerning the reference has been obtained from Equation (2). The abovementioned data is shown in Table 4.

The mean value recorded by the reference device during the exposure period is around 625 Bq/m^3^, with a median of 241 Bq/m^3^ and a maximum value of 14,687 Bq/m^3^. The EcoQube, RadonEye and RadonEye Plus2 devices are found to reach the maximums specified by the manufacturer (see Table 1) at 3676, 3685 and 9418 Bq/m^3^, respectively. In the other cases, the maximum value recorded is always below the reference value, slightly lower for the Spirit monitor and almost half that for the ViewPlus, ViewRadon and WavePlus monitors.

A comparison of the mean and the median suggests that the data do not follow a normal distribution due to a difference of more than double in all cases. The mean is biased by the higher extreme values. Therefore, the dispersion of the data obtained from the interquartile range (IQR) shows a high dispersion of the data. The difference between the mean value measured by each monitor and the mean value measured by the reference monitor does not differ in any case by more than 19%, despite the loss of data and the limitation of some of the monitors in terms of measurement range, as shown above.

According to the evolution of radon concentration over time recorded by the monitors and the corresponding to the reference are similar as shown in Figure 3, the observed *D* differences are not very large, ranging from approximately 1% to 19% (see Table 4). In order to mathematically determine whether the evolution of each monitor follows the reference, and to assess whether the differences are not significant, the Pearson correlation coefficient and the *t*-test results were obtained, respectively. As the results of Table 5 show, it is confirmed that in all cases, there is a good correlation between the evolution shown by the reference monitor and that provided by the monitors studied; in all cases, the correlation is significant, and the correlation coefficient *r* is higher than 0.70. The *p*-value of Pearson’s test has been defaulted to <0.0001 because the results obtained are less than 10^−140^ in all cases.

In the last two weeks of the exposure period, daily variations in radon concentration are clearly observed (see Figure 4). Moreover, the levels are not as high, the average value measured by the reference is 210 Bq/m^3^, so this kind of behaviour could be more similar to an occupational or household situation. In the same way as for the entire exposure period, each device’s average, the median, the maximum value reached, the interquartile range IQR, and the percentage difference concerning the reference, has been obtained for the two last weeks. Results are shown in Table 6.

It is observed that most of the devices describe the daily variations, reaching in most cases the AlphaGUARD maximums; however, in many other cases, the lower values are not measured. This causes all monitors to overestimate the radon concentration. Although the percentage difference *D* (%) is high, the absolute difference is not much, between 10 and 80 Bq/m^3^.

The study of the response time *t_r_* of the devices has been carried out based on the abrupt variation in the radon concentration within the general evolution over the whole period of Figure 3. We took a symmetrical period in which data from all monitors were available to evaluate both the rise and fall after reaching the maximum concentration, and that there were no limitations in the range of concentrations, at least to reach the percentage of the maximum level required. The first selected period that meets these requirements is 8–9 July, where the initial radon concentration measured by the AlphaGUARD is 400 Bq/m^3^, with a maximum peak of 3800 Bq/m^3^.

Figure 5 shows the time evolution of all monitors. The start of the peak is indicated with a red arrow, the 10%, 50% and 90% of the absolute variation over the initial value are indicated with black horizontal lines, corresponding to the values 740, 2100 and 3460 Bq/m^3^, respectively. Table 7 specifies the time each monitor takes to reach the above concentrations in both increase and decrease from the start.

The fastest responding device is the AlphaGUARD, followed by the Spirit and the EcoQube in roughly equal measure, despite the two having different integration times. The analysis of the RadonEye devices reveals that the enhanced version (RadonEye plus2) responds 1 h faster in all cases than the standard version and 3 h slower on average than the reference monitor. ViewPlus, ViewRadon and WavePlus only reach the 10% level, failing to reach the 50% and 90% levels. The evolution of a peak with an absolute increase of 3400 Bq/m^3^ that has taken 8.16 h for the reference monitor is too fast for these devices, which cannot follow such an evolution, taking more than 25 h.

The second period used to study the response time was a symmetrical interval as in the case studied above (see Figure 5), but this time for a peak obtained from the period of Figure 4, where the variation in concentration is lower in approximately the same amount of time. The dates chosen were between 13 and 14 August, the evolution of which can be seen in Figure 6, where the initial radon concentration measured by the AlphaGUARD is 179 Bq/m^3^, with a maximum peak of 1048 Bq/m^3^. Table 8 specifies the time each monitor takes to reach the above concentrations in both increase and decrease from the start.

The fastest responding devices again are the AlphaGUARD, the EcoQube and the Spirit. In the increase area, the best performances are distributed; however, in the decrease area the one who responds best seem to be the AlphaGUARD, followed by the Spirit and the EcoQube. For these radon concentration levels, the RadonEye and RadonEye Plus2 devices respond in a very similar way. An average delay of 2.5 h is found with respect to the fastest devices. They failed to reach the 90% level in any case. As in Period (1), ViewPlus, ViewRadon and WavePlus devices only manage to reach the level corresponding to 10% of the total increase. Because there are successively more daily period variations, these devices do not manage to go below 10% again.

## 4. Conclusions

In this study, a comparison of low-cost radon monitors has been carried out at the Laboratory of Natural Radiation (LNR) between 1 July 2023 and 23 August 2023 under field conditions. Seven monitors with different characteristics regarding detection technology, measurement range, sensitivity and uncertainty have been evaluated. The reference value of radon concentration was obtained from an AlphaGUARD monitor calibrated at the Laboratory of Environmental Radioactivity, University of Cantabria (LaRUC), which is accredited for testing and calibration according to ISO/IEC 17025 [16].

The background of the studied monitors has been determined to be below 20 Bq/m^3^ in all cases, which, although it is necessary to subtract from the raw measurement, the error made will depend on the radon atmosphere in which one is working. This data will be relevant when measuring low radon concentrations, and it would not be relevant in the case of the study described in this article if no such consideration had been made.

In addition, temporal stability has been studied based on the lack of registration in the time series. In all cases, there is a percentage of failure ranging from 1% to approximately 19% of the data.

The main technical characteristics studied in relation to the measurement of radon concentration were measurement ranges, accuracy, correlation, and response time. Based on the results obtained, we can determine that the measurement ranges are in accordance with those specified by the manufacturers, with three of the seven monitors exceeding their maximum established range. The percentage difference concerning the reference monitor is between 5% and 16% in all cases for the entire exposure period, with the differences being non-significant according to the contrast hypothesis test applied, except for one of the monitors. The correlation obtained in all cases is significant, with a Pearson correlation coefficient between 0.63 and 0.90. Over the period considered of the last two weeks of exposure where radon variations are smaller, larger percentage differences have been found over the last two weeks of exposure.

The monitors’ response is variable in time depending on the period considered, with delays between 1 and 3 h from the reference for the EcoQube, RadonEye, RadonEye Plus2 and Spirit monitors. ViewPlus, ViewRadon, and WavePlus devices stay within the 10% level, with differences in response times between 5 and 15 h. In the latter cases, the correlation is the lowest obtained due to the time lag between the series of the abovementioned monitors and that of AlphaGUARD.

The response time derived from the analysis may suggest that reducing the integration time of some monitors would improve the response. However, the response time is related to the sensitivity of the equipment. In the case of high radon concentrations, such as those generally measured in this study, there is sufficient count statistic to reduce the integration time and thus improve the response time. However, at environmental levels for which the equipment is designed, longer integration times are required because the uncertainty would increase too much due to the low count, and a lot of information would be lost due to the dispersion of the equipment itself.

Although the general purpose of these monitors is for non-professional use, overall, the results show that monitors under study provide acceptable radon time series for their intended purpose, the measurement of indoor radon at a user level, despite the monitors’ data loss, response time and limited maximum measurement range. This type of device can make acceptable estimates of average radon concentration values for long periods of exposure, which is undoubtedly helpful in radiation protection against radon. However, their response may be compromised when radon concentration levels are very high or highly variable, as may be true for specific jobs with more demanding technical requirements, such as mitigation or research studies.

An additional study linked to the one proposed in this article would be the experimental determination of the accuracy claimed by the manufacturer, which requires stable radon concentration levels and the evaluation of the dispersion of the measurements, considering the specified integration times. In addition, apart from the radon findings, the other environmental and air quality parameters measured by the devices can determine the suitability of the devices. However, these parameters have not been considered in this study.

## Figures and Tables

**Figure 1 sensors-24-01836-f001:**
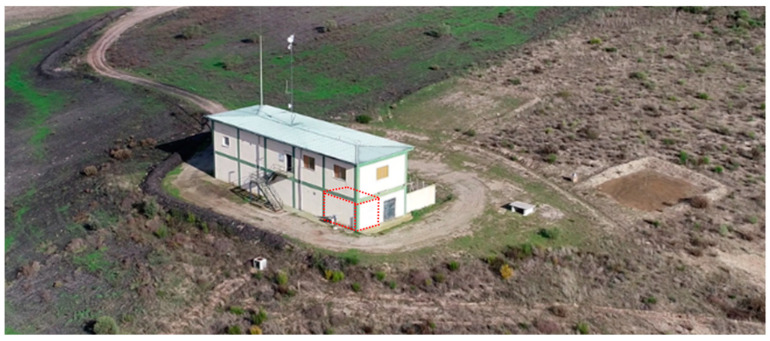
Aerial view of the LNR showing Room1 inside the building marked in red.

**Figure 2 sensors-24-01836-f002:**
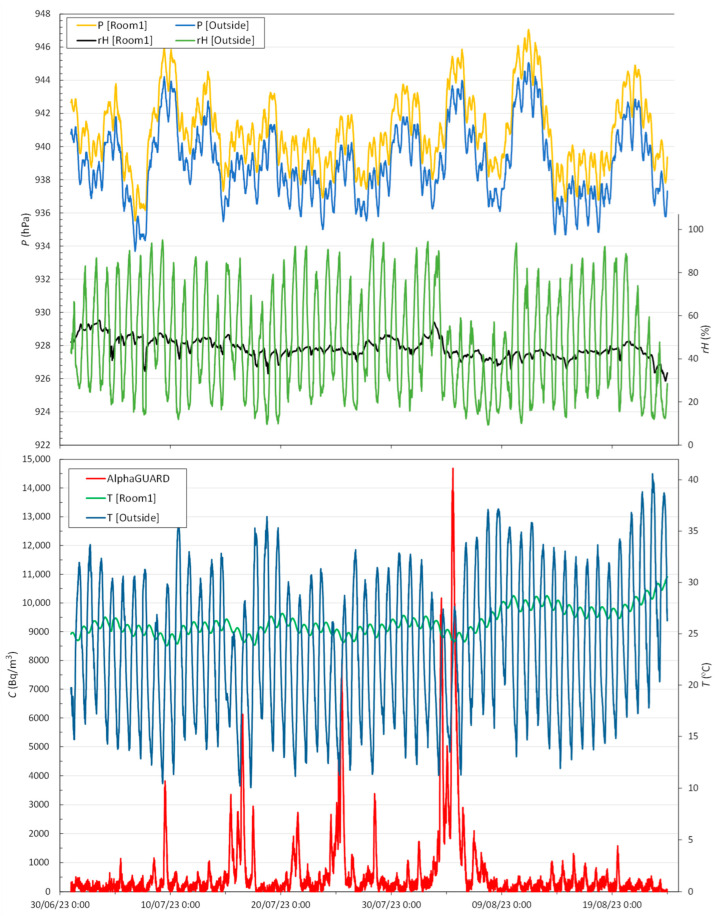
Radon concentration in Room1 measured by the reference device and the environmental conditions (atmospheric pressure *P*, relative humidity *rH* and temperature *T*) inside and outside every 10 min.

**Figure 3 sensors-24-01836-f003:**
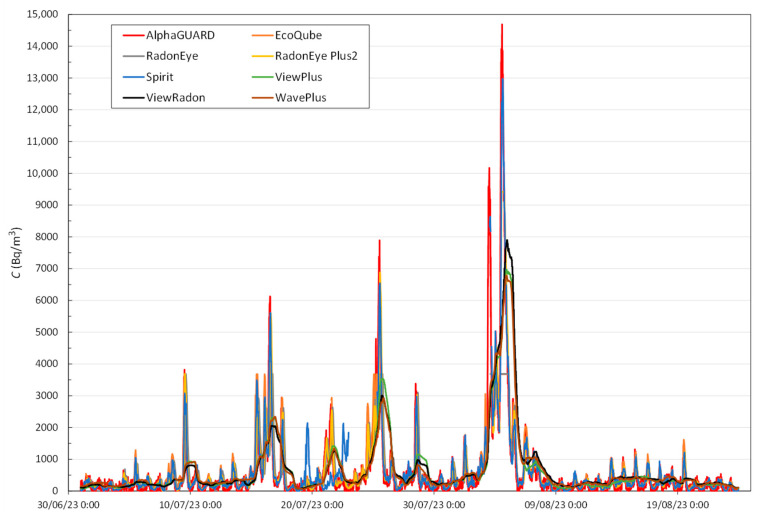
Radon concentration in Room1 measured by the reference device and the specified monitors.

**Figure 4 sensors-24-01836-f004:**
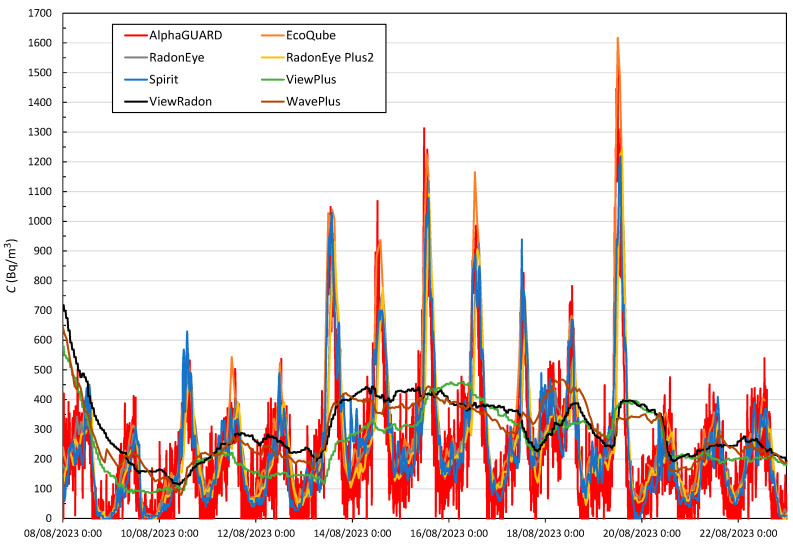
Radon concentration in Room1 measured by the reference device and the monitors under study from 8 August 2023 to 23 August 2023.

**Figure 5 sensors-24-01836-f005:**
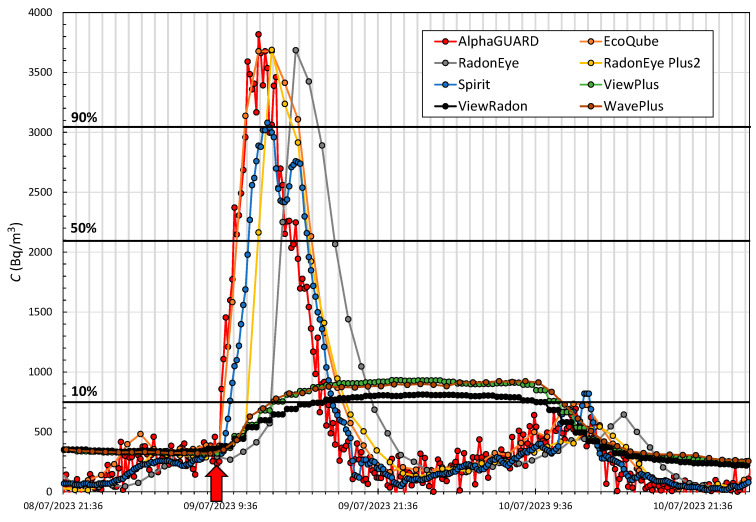
Period (1) of evolution of the radon concentration in Room1 measured by the reference device and monitors to study the response time. A red arrow indicates the start considered, and horizontal black lines indicate 10%, 50% and 90% of the net value of the peak.

**Figure 6 sensors-24-01836-f006:**
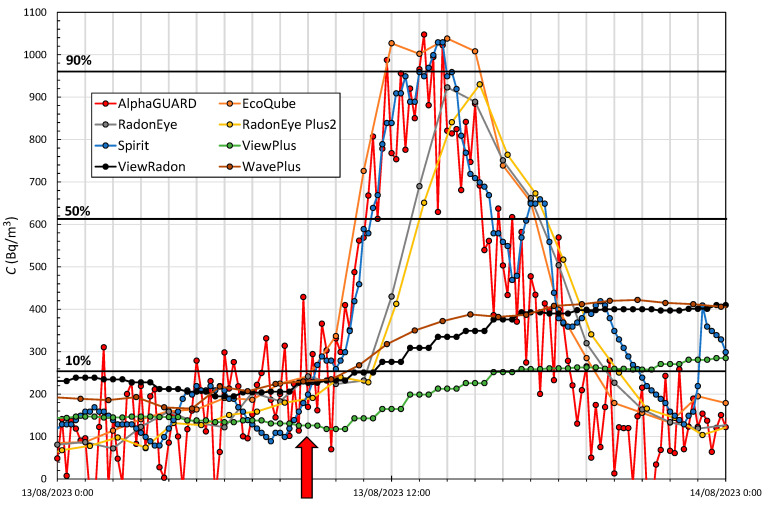
Period (2) of evolution of the radon concentration in Room1 measured by the reference device and monitors to study the response time. A red arrow indicates the start considered, and horizontal black lines indicate 10%, 50% and 90% of the net value of the peak.

**Table 1 sensors-24-01836-t001:** Technical characteristics of the radon monitors used in the study relating to radon measurement given by the manufacturer.

Device	Manufacturer	Detection Principle	Range (Bq/m^3^)	Sensitivity (cpm at 1 kBq/m^3^)	Radon Uncertainty (Statistical)
AlphaGUARD ^1^	Bertin Instruments (Frankfurt, Germany)	Pulsed Ionization chamber	2–2 M	50	<±10% at 200 Bq/m^3^ after 1 h
EcoQube	Ecosense (California, USA)	Pulsed Ion Chamber	7–3700	14	<±10% at 370 Bq/m^3^ after 10 h
RadonEye	FTLab (Asan, South Korea)	Pulsed Ion Chamber	7–3700	14	<±10% at 370 Bq/m^3^ after 10 h
RadonEye Plus2			7–9435		
Spirit	Radonova (Uppsala, Sweden)	Semiconductor detector with alpha spectroscopy	0–100 k	40	<±15% at 200 Bq/m^3^ after 6 h
ViewPlus	Airthings (Oslo, Norway)	Photo diode Alpha spectrometry	0–20 k	no info.	<±10% at 200 Bq/m^3^ after 7 days
ViewRadon					
WavePlus					

^1^ Used as standard reference.

**Table 2 sensors-24-01836-t002:** Background *B* obtained for each of the monitors, its standard deviation SD and used integration time Δ*t*.

Device	*B* (Bq/m^3^)	SD (*B*) (Bq/m^3^)	Δ*t* (min)
AlphaGUARD	103	68	10
EcoQube	20	10	60
RadonEye	11	7	60
RadonEye Plus2	17	9	60
Spirit	21	13	10
ViewPlus	18	2	20
ViewRadon	0	0	20
WavePlus	0	0	60

**Table 3 sensors-24-01836-t003:** Mean, standard deviation (SD), minimum value (Min.), and maximum value (Max.) of the environmental conditions in Room1 and outside during the exposure period.

Parameter	Mean	SD	Min.	Max.
*T* (°C) [Room1]	26.1	1.3	23.8	30.5
*rH* (%) [Room1]	44.9	4.6	29.7	57.9
*P* (hPa) [Room1]	940.9	2.2	935.5	947.1
*T* (°C) [Outside]	23.6	6.7	10.0	40.6
*rH* (%) [Outside]	46.9	23.0	9.2	95.7
*P* (hPa) [Outside]	939.0	2.2	933.7	945.1

**Table 4 sensors-24-01836-t004:** Mean value, median, interquartile range IQR (Q_3_-Q_1_), maximum value (Max.), percentage of non-registered data (Data lost) and percentage difference (*D*) over the entire exposure period.

Device	Mean (Bq/m^3^)	Median (Bq/m^3^)	IQR (Q_3_−Q_1_) (Bq/m^3^)	Max. (Bq/m^3^)	Data Lost (%)	*D* (%)
AlphaGUARD	625	241	379	14,687	0.0	-
EcoQube	659	291	464	3676	18.9	5.4
RadonEye	526	257	332	3685	2.2	−15.8
RadonEye Plus2	566	243	308	9418	2.4	−9.4
Spirit	594	239	370	12,979	6.4	−4.9
ViewPlus	673	294	414	6965	1.6	7.7
ViewRadon	687	322	389	7900	0.8	10.0
WavePlus	694	343	397	6782	6.1	11.0

**Table 5 sensors-24-01836-t005:** Results of the Student’s *t*-test and Pearson’s correlation coefficient *r*, indicating the *p*-value in each case, whether the means can be considered equal and whether there is a correlation between the series of each monitor and that of the reference.

Device	t-Statistic	t-Statistic (*p*-Value)	Equal Means?	*r*	*r* (*p*-Value)	Correlation?
EcoQube	−0.70	0.50	yes	0.78	<0.0001	yes
RadonEye	2.27	0.02	no	0.70	<0.0001	yes
RadonEye Plus2	1.21	0.23	yes	0.87	<0.0001	yes
Spirit	0.41	0.68	yes	0.90	<0.0001	yes
ViewPlus	−0.96	0.34	yes	0.64	<0.0001	yes
ViewRadon	−1.24	0.21	yes	0.63	<0.0001	yes
WavePlus	−1.48	0.14	yes	0.69	<0.0001	yes

**Table 6 sensors-24-01836-t006:** Mean value, median, interquartile range IQR (Q_3_−Q_1_), maximum value (Max.) and percentage difference (*D*) over the last two weeks of exposure from 8 August 2023 to 23 August 2023.

Device	Mean (Bq/m^3^)	Median (Bq/m^3^)	IQR (Q_3_−Q_1_) (Bq/m^3^)	Max. (Bq/m^3^)	*D* (%)
AlphaGUARD	210	169	418	1580	-
EcoQube	274	217	215	1617	30.9
RadonEye	236	205	177	1149	12.7
RadonEye Plus2	221	184	177	1241	5.6
Spirit	232	189	210	1219	10.9
ViewPlus	249	217	161	579	18.6
ViewRadon	293	268	160	718	39.9
WavePlus	281	261	170	621	34.1

**Table 7 sensors-24-01836-t007:** Response time *t_r_* indicated in hours for each monitor to reach the specified radon concentration values of 10%, 50% and 90% in both increase and decrease for Period (1).

	*t_r_* (Increasing) (h)			*t_r_* (Decreasing) (h)		
Device	10%	50%	90%	90%	50%	10%
AlphaGUARD	0.16	1.16	2.16	4.5	6	8.16
EcoQube	1	2	2	7	8	10
RadonEye	5	5	6	8	9	12
RadonEye Plus2	4	4	5	7	8	11
Spirit	1.16	2.66	4	4.33	7.33	9.16
ViewPlus	5.66	-	-	-	-	26.33
ViewRadon	8.33	-	-	-	-	25.33
WavePlus	5	-	-	-	-	28

**Table 8 sensors-24-01836-t008:** Response time *t_r_* indicated in hours for each monitor to reach the specified radon concentration values of 10%, 50% and 90% in both increase and decrease for Period (2).

	*t_r_* (Increasing) (h)			*t_r_* (Decreasing) (h)		
Device	10%	50%	90%	90%	50%	10%
AlphaGUARD	1	2.16	4	5	6.33	9.33
EcoQube	1	2	3	7	10	13
RadonEye	3	4	-	-	9	11
RadonEye Plus2	3	4	-	-	9	11
Spirit	0.33	2.33	4.33	5	8.66	12
ViewPlus	7	-	-	-	-	-
ViewRadon	3	-	-	-	-	-
WavePlus	2	-	-	-	-	-

## Data Availability

The data presented in this study are available in Appendix A.

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
