# Peer review of "Investigation of the Performance of Various Low-Cost Radon Monitors under Variable Environmental Conditions"

_sensors, 2024, doi:10.3390/s24061836_

Round 1

Reviewer 1 Report

Comments and Suggestions for Authors 1. The research is evaluating the efficiency of 7 radon indoor monitoring devices coming from 6 producers.  The tested monitoring devices worked on 3 different principles of measurement: Pulsed ion Chamber, semiconductor alpha spectrometry and photo-diode alpha spectrometry.  2.  Give the significant public health and occupational safety based on radon inhalation associated with increased health risk of lung cancer due to alpha radon decay, the research is relevant because it only considers low-cost detectors affordable by small businesses or individuals. 3.It is a study only on affordable measuring devices and comes to confirm the effectiveness indicated by the manufacturer. 4. The author should consider real case situations, putting the detectors in real conditions, area with people and air flow. 5.The main technical characteristics: accuracy, correlation and response time, determined the measurement range according to the manufacturer's specifications in a range below the significant threshold. 6. References are adequate even if they can be improved. 7.Grayscale chart representation is a bit difficult to follow, so colored editing may be an improvement.  

raw 196 check the editing way

Reviewer 2 Report

Comments and Suggestions for Authors

See attached PDF.

Comments on the Quality of English Language

Very few - see attached PDF.

Reviewer 3 Report

Comments and Suggestions for Authors

A nice work with valuable refenence for readers

Mu comment and Suggestions:

1. Keywords: "low-cost radon monitor" should be added to.

2. Table 1. name of monitors, like in the abstract and Table 2, should be given for the consistence and better understand for readers.

3. “Low-cost radon monitor” is not a good classification name. For this study, passive monitors with data storage capacity of each measurement interval were selected. So if possible, try to consider another name rather than low-cost monitor.  

4. For discussion on response time of monitors, attention should be paid to the different specific purpose of development of each monitor, which relays on the different interval time, some are 60 min and some are 10min or 20 min.  

Reviewer 4 Report

Comments and Suggestions for Authors

Radon gas is a radioactive gas commonly found in the natural environment, and excessive inhalation can pose significant health risks to the human body. The measurement of radon concentration is a crucial initial step in radon gas protection.

General comment:

This paper compares several low-cost gas radon detectors and compares their measurement range, measurement accuracy, and response time information, which is a guide for the selection of gas radon concentration meters. However, this paper is basically not innovative and is just a routine test of commercial instruments. It does not meet the journal's publication requirements, which can be found at Sensors | Special Issue : Detection and Measurement of Radioactive Noble Gases (mdpi.com). Therefore, I have to recommend rejection.

Specific comments:

1. Lines 69-72: Given that the test was conducted in this room, it is advisable to furnish additional details about the room, at least the radon source and the stability of radon concentration within the room should be included.

2. Table 1.

Sensitivity (cpm at 11 kBq/m3)

I don't think it's appropriate to refer to this parameter as sensitivity, it should be called calibration factor because the parameter converts the measured event rate to radon concentration.

 About the detection principle

Does “Pulsed Ion Chamber” have the same meaning as “Pulsed Ionization chamber”? If not, what is the difference? If yes, why do you use different descriptions?

For “Spirit, ViewPlus, ViewRadon, WavePlus”, they could measure the alpha spectrum, then which alpha is used to determine the radon concentration? There are several alpha sources in the radon decay chain.

3. Line 118,

for F=1.00 +- 0.09, how is the error derived?

radon C concentration ->  radon concentration C

4. Lines 121-122

According to the definition of F, I think it should be called “correction  factor”.

Dese the “F” here the same as presented in line118? It's a little strange to see two definitions.

5. Table 2:

How is SD defined? I think SD needs to be dependent on the distribution of B.

6. Lines 177-178:

I would suggest adding a curve of radon concentration changes in a day to illustrate this.

7. Lines 188-193:

At what radon concentration were these counts obtained, given that the change in radon concentration varied by four orders of magnitude?

8. Lines Line 197-199:

The standard deviation significantly exceeds the central value, and it is recommended that the authors revise the expression.

9.Fig.3:

It appears that A and B are the same color, therefore, for comparison purposes, it is recommended to change the color of one of them.

The measurement results of ViewPlus, ViewRadon, and WavePlusABC shown in this figure are significantly different from the reference and the others, does this mean that ABC's measurements are not reliable?

Comments on the Quality of English Language

The English looks good to me. 

Round 2

Reviewer 4 Report

Comments and Suggestions for Authors

This paper has been improved and the authors have addressed all my comments.